# Decoupling Noise and Toxic Parameters
# for Language Model Detoxification by Task Vector Merging

**Yongmin Kim[1]**    **Takeshi Kojima[1]**    **Yusuke Iwasawa[1]**    **Yutaka Matsuo[1]**
[1]The University of Tokyo
kimyongmin@robot.t.u-tokyo.ac.jp,
{t.kojima, iwasawa, matsuo}@weblab.t.u-tokyo.ac.jp

## Abstract

The goal of detoxifying language models is to reduce the chances of produc-
ing offensive or harmful output in pre-trained language models (PLMs),
ensuring their safer use. A recently proposed detoxification method utilizes
the task vector obtained by subtraction from the fine-tuned model on toxic
datasets to the pre-trained model. This approach has shown effectiveness
for detoxification but still suffers from degradation. This study focuses
on further mitigating degradation while maintaining detoxification perfor-
mance. To mitigate the degradation, we propose a method that detoxifies
the PLMs by fine-tuning multiple models on split toxic datasets and by
merging the subtracted task vectors. We conducted experiments on two
toxic datasets (Civil Comments and Toxigen) with five PLMs (GPT2-small,
GPT2-medium, GPT2-large, Phi-1.5, and Llama2-7b), demonstrating that
our method consistently achieves a lower toxicity score while preventing
the degradation compared to baseline methods. Especially, with the GPT2-
small model on the Toxigen dataset, degradation was reduced by 38.9%
compared to that of an existing task vector method while maintaining a
similar toxicity score. In addition, we found that merging multiple detox-
ified models tends to increase the number of parameters that remained
almost unchanged from the pre-trained model. We assume that by merging
multiple detoxified models, "decoupling noise and toxic parameters" is
implicitly achieved. The accidental noise in the parameter shift unrelated to
detoxification disappears by averaging noise, whereas the parameter shift
associated with detoxification is maintained. We hope that the findings of
this study will be applied not only to detoxification but also to many other
research domains that seek to suppress undesirable outputs of language
models. [1]

## 1    Introduction

Although pre-trained language models (PLMs) have shown remarkable capabilities across
many domains over the past few years (Radford et al., 2019; Brown et al., 2020; Li et al.,
2023; Touvron et al., 2023), PLMs have the risk of generating offensive and aggressive
contents (Gehman et al., 2020; Liang et al., 2023). Toxic data present in pretraining datasets
causes PLMs to generate rude and biased sentences. Recently, to enhance the secure
utilization of LMs, the crucial challenge of detoxifying LMs has attracted growing research
interest (Kumar et al., 2023).

To mitigate the risk and provide safer and non-aggressive PLMs, detoxification methods that
prevent the generation of toxic sentences have been gaining attention. Previous studies (Liu
et al., 2021; Krause et al., 2021; Kwak et al., 2023) proposed detoxification methods that
adjusted the output probability of the next token to prevent potentially toxic tokens. Other

---

[1]Code is available at https://github.com/oishikimchi97/merge_to_detoxify
**Warning: this paper includes rude or offensive examples in Appendix.**

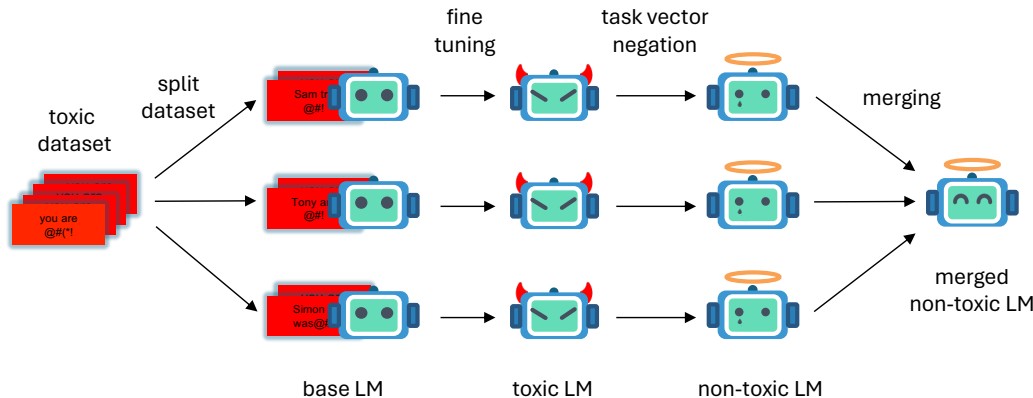

Figure 1: Overview of our method. We split a whole toxic dataset into $N$ sub-datasets. Then, we fine-tune the base-LM on each sub-dataset. We detoxify each model using the task vector negation. We then merge these detoxified models into one model to mitigate the degradation.

studies (Gururangan et al., 2020; Wang et al., 2022; Lu et al., 2022) proposed methods of reducing the toxicity of the model through post-training.

Recently, Ilharco et al. (2023) proposed another method, a task vector operation, which achieves detoxification from a toxic dataset with arithmetic operations on the model's parameters. The task vector is the difference in the parameter space between a fine-tuned model and a pre-trained model. They have shown that simple arithmetic operations, such as subtracting the task vector to a pre-trained model, can effectively unlearn undesirable attributes, such as toxicity. However, as other methods, this detoxification method degrades the existing LMs' ability. A trade-off relationship exists between the detoxification performance and the degradation of LMs' capabilities, which needs improvement.

Inspired by the recent success of the model merging technique Gueta et al. (2023); Wortsman et al. (2022), which merges multiple model parameters to achieve higher performance, we propose a novel method that merges multiple task vectors to mitigate degradation further and boost the detoxification performance. Specifically, we randomly split a whole toxic dataset into $N$ sub-datasets. Then, we fine-tune the base-LM on each sub-dataset. We detoxify each fine-tuned model using task vector negation and merge the N detoxified models into one model (Figure 1 presents an overview of our method).

To validate the effectiveness of our method, we conducted experiments on two toxic datasets (Civil Comments and Toxigen) and five PLMs (GPT-2, GPT-2 medium, GPT-2 large, Phi-1.5, and Llama2-7b). Our experiment results demonstrate that our proposed method consistently achieves higher detoxification while suppressing degradation compared to baseline methods, including the standard task vector negation approach. We found that the individual models detoxified from each sub-dataset have much less degradation than a model detoxified from the entire dataset. We also found that the merged models showed a similar level of toxicity with less degradation compared to the individual models detoxified from sub-datasets.

Furthermore, to investigate the root cause of the degradation mitigation achieved by our method, we measured the similarity between our detoxified models and the pre-trained model in the parameter space. We found that merging multiple detoxified models tends to increase the number of parameters that remained almost unchanged from the pre-trained model. We assume that by merging multiple detoxified models, "decoupling noise and toxic parameters" is implicitly achieved. The accidental noise in parameter shift unrelated to detoxification disappears by noise averaging, while the parameter shift associated with detoxification is maintained.

## 2 Related work

### 2.1 Detoxification

The ability of PLMs to generate text across various contexts is derived from their pre-training on large, multi-domain textual datasets (Prabhumoye et al., 2023; Korbak et al., 2023). Nonetheless, given that these datasets are scraped from web content, they unavoidably include harmful material (Gehman et al., 2020; Gao et al., 2020; Penedo et al., 2023; Kumar et al., 2023), thus increasing the likelihood of PLMs producing toxic outputs. The objective of language detoxification is to minimize the chances of producing continuations that contain toxic elements. Toxic elements are defined as aspects of the text likely to be perceived as impolite, disrespectful, or offensive (Kwak et al., 2023; Schick et al., 2021).

Recent studies address language model toxicity by aligning outputs with human preferences (Radford et al., 2019), modifying decoding to suppress toxic tokens (Liu et al., 2021; Krause et al., 2021; Kwak et al., 2023) or exploring detoxification through toxic prompts (Leong et al., 2023). However, these methods need additional inference time. In contrast to these approaches, certain research works (Gururangan et al., 2020; Wang et al., 2022; Lu et al., 2022; Tang et al., 2024) have proposed methods to fine-tune non-toxic data. In line with these methods, Ilharco et al. (2023) has demonstrated the effectiveness of the task vector negation method to detoxify LMs. The task vector approach can utilize only toxic data for detoxification as opposed to the previous trainable methods.

Ilharco et al. (2023) introduced the concept of the task vector, demonstrating that through arithmetic operations on the task vector, one can enhance the model's performance, unlearn an undesired attribute, or improve domain generalization. Their study proposed a method of detoxification using the task vector from a model fine-tuned on a toxic dataset. The detoxification performance of this method was better than that of methods fine-tuned on non-toxic datasets and it led to less degradation compared to that of methods using the gradient ascent. In contrast to the existing detoxification methods, this approach uses only toxic datasets for detoxification.

However, this approach for detoxification weakens the inherent abilities of LMs, highlighting a trade-off between detoxification efficacy and degradation of a model's performance. We have demonstrated that, compared to the existing task vector method, our approach further mitigates the degradation from the detoxification at the same toxicity. Furthermore, our method is cost-effective compared to other trainable methods(Gururangan et al., 2020; Wang et al., 2022; Lu et al., 2022; Tang et al., 2024) because it uses only toxic data.

### 2.2 Model merging

Recent studies(Gueta et al., 2023; Wortsman et al., 2022) have shown that merging the weights of different models can yield a model with better performance than that achievable with trainable methods. Wortsman et al. (2022) proposed a method to efficiently merge different models linearly by selecting models from candidates. Furthermore, Matena & Raffel (2022); Daheim et al. (2024b); Yadav et al. (2023) proposed more computationally efficient and effective methods for fusing multiple models to enhance performance. In this study, we adopt the linear merging method and additionally apply Tie-Merging(Yadav et al., 2023) to the proposed method.

## 3 Method

### 3.1 Preliminary

Ilharco et al. (2023) proposed a method for detoxification by negating the task vector calculated with a model fine-tuned on a toxic dataset. This method proved to be more effective in detoxification and less prone to performance degradation compared to methods fine-tuning on non-toxic datasets or employing the gradient ascent on toxic data.

Suppose that $\theta_{\mathrm{pre}} \in \mathbb{R}^d$ represents the pre-trained model's parameters, and $\theta_{\mathrm{ft}}^t \in \mathbb{R}^d$ is the parameter after fine-tuning for a particular task $t$. The task vector, denoted as $\tau_t \in \mathbb{R}^d$, is calculated by the element-wise subtraction of $\theta_{\mathrm{pre}}$ from $\theta_{t_{\mathrm{ft}}}$, expressed as $\tau_t = \theta_{\mathrm{ft}}^t - \theta_{\mathrm{pre}}$. If the task is apparent, we omit the identifier $t$ and simply notate it as $\tau$. These task vectors can be utilized across any model parameters $\theta$ from an identical architecture by adding them element-wise with a scaling coefficient $\lambda$, leading to a new model parameter $\theta_{\mathrm{new}} = \theta + \lambda\tau$. Negating the task vector $\tau$ by applying the inverse vector $\tau_{\mathrm{new}} = -\tau$ extends the fine-tuned model back towards the pre-trained model. This operation leads to a diminished or unlearned model's ability to perform the specific task for which it was fine-tuned (e.g., toxicity).

## 3.2  Proposal

Inspired by the recent success of model merging techniques (Gueta et al., 2023; Wortsman et al., 2022), which merge multiple model parameters to achieve higher performance, we propose a novel method of merging multiple task vectors to mitigate degradation further and enhance detoxification. Specifically, we divide the dataset $\mathcal{D}$ into $N$ sub-datasets $\mathcal{D}_{(\mathrm{N,i})}$, ensuring that no common parts exist among them, which can be mathematically represented as follows:

$$\bigcup_{i=1}^{N} D_{(\mathrm{N,i})} = D. \tag{1}$$

Zaman et al. (2023) demonstrated that when fusing several models, the task abilities of each model are reinforced in their common aspects and suppressed in their differing aspects after merging. Furthermore, they showed that merged models have an improved generalization ability. Based on these findings, we hypothesized that merging models fine-tuned on each sub-dataset would result in a suppression of the degradation, an unshared attribute across the models. Ultimately, we detoxify the pre-trained model by negating the task vector of merged models. Although many methods exist for merging models (Yadav et al., 2023; Daheim et al., 2024b; Matena & Raffel, 2022), we adopted a simple approach to clarify the effectiveness of our method. Specifically, we considered merging by computing the weighted average of the model parameters. Let there be $N$ models that are fine-tuned to each subset of a toxic dataset $\mathcal{D}_{(\mathrm{N,i})}$ with parameters $\{\theta_1 \ldots \theta_N\}$ such that $\forall_i \theta_i \in \mathbb{R}^d$. Then, we define the merged model, $\theta_{\mathrm{merged}}$, as a convex combination:

$$\theta_{\mathrm{merged}} = \sum_{i=1}^{N} \alpha_i \theta_i \tag{2}$$

where $\alpha_i \geq 0, \sum_{i=1}^{N} \alpha_i = 1$. We employ the average for weight merging, where $\alpha_1 = \alpha_2 = \cdots = \alpha_N = 1/N$. From the merged model $\theta_{\mathrm{merged}}$, we can calculate the task vector $\tau_{\mathrm{merged}} = \theta_{\mathrm{merged}} - \theta_{\mathrm{pre}}$. Consequently, we can obtain the detoxified model $\theta_{\mathrm{detoxic}}$ by the negation of the task vector $\tau_{\mathrm{merged}}$ to the pretrained model $\theta_{\mathrm{pre}}$ with coefficient $\lambda$. To be specific, we compute the detoxified model with Equation (3).

$$\theta_{\mathrm{detoxic}} = \theta_{\mathrm{pre}} - \lambda\tau_{\mathrm{merged}}. \tag{3}$$

## 4  Experiment setting

In this study, we detoxified the models of GPT2 (Radford et al., 2019), Llama2-7b, and Phi-1.5 (Li et al., 2023) using a toxic dataset included in the dataset. For the proposed method, we divided the entire toxic dataset into sub-datasets and fine-tuned the models on them. Then, we merged those models and calculated the task vector from it. Finally, we detoxified the model with the negation of the task vector. To verify the proposed method in diverse settings, we conducted experiments through multiple datasets for toxic data.

### 4.1 Datasets

We employed Civil Comments(Borkan et al., 2019) and Toxigen(Hartvigsen et al., 2022) as toxic datasets for detoxification. We made use of highly toxic data included in these datasets. For the toxicity evaluation, we used prompts in RealToxicPrompts(Gehman et al., 2020) and Toxigen(Hartvigsen et al., 2022). We sampled 1K prompts from these datasets. We provide more detailed information about the datasets in Section C.1

### 4.2 Baselines

For the baselines, we used a set-up from Ilharco et al. (2023) for fine-tuning on the non-toxic dataset, and trained the model to the toxic dataset with the gradient ascent approach (Tarun et al., 2021; Jang et al., 2023). We fine-tuned the models to the non-toxic data (toxicity score lower than 0.2) included in each dataset, in a similar way to Liu et al. (2021). We did not use the entire non-toxic dataset; instead, we sampled the same number of non-toxic data from the entire toxic dataset as the toxic data we used. In the detoxification experiments using gradient ascent, we maximized the negative log-likelihood loss on the toxic dataset. Finally, we compared the proposed method with detoxification using an entire toxic dataset realized by Ilharco et al. (2023).

We did not conduct a comparison between our proposed method and existing self-detoxification approaches (Wang et al., 2022; Tang et al., 2024). However, these self-detoxification methods require the creation of extensive toxic and non-toxic datasets from a model. These datasets need to be generated for each model, and thus, a large amount of data are required for each detoxification. By contrast, the negation method adopted in the proposed approach allows for efficient detoxification using only existing toxic datasets

### 4.3 Experiment

First, we examine the impact of the proposed method on different sizes of sub-datasets $N$ in Section 5.1. Second, we investigate the effectiveness of merging models detoxified from the split datasets in Section 5.2. Third, we compare the proposed method with the existing task vector negation approach by examining the parameter space of the detoxification models in Section 5.3. Specifically, we measure how different the parameters of detoxification models, which have the same toxicity level, are from the pre-trained model. Finally, we compare our approach with the baseline methods in Section 5.4.

### 4.4 Evaluation

In each experiment, we measured the toxicity and degradation of the models to compare the baseline methods and our approach. According to Gehman et al. (2020), a toxic sentence is defined as "a rude, disrespectful, or unreasonable comment that is likely to make you leave a discussion". Toxicity is evaluated through the toxicity score measured by pre-trained toxicity detectors (Hanu & Unitary team, 2020; Hartvigsen et al., 2022) and Perspective API[2], which is widely used as a toxicity detector. Different toxic detectors were employed depending on the experiment. Detailed information can be found in each experiment section. In Section 5.1, 5.2, 5.3, we assess the model's degradation through perplexity. In Section 5.4, we also measure the average performance of models on seven downstream tasks to comprehensively evaluate the models' degradation, in addition to assessing perplexity.

## 5 Experiment results and discussion

### 5.1 Impact of the sub-dataset size

**An increase in the number of sub-datasets mitigates degradation while maintaining a similar toxicity level.** To estimate the impact of the number of sub-datasets $N$ on detoxification, we conducted the experiments by setting $N$ to $3, 5$, and $10$. Then, we compared those

---

[2]`https://github.com/conversationai/perspectiveapi`

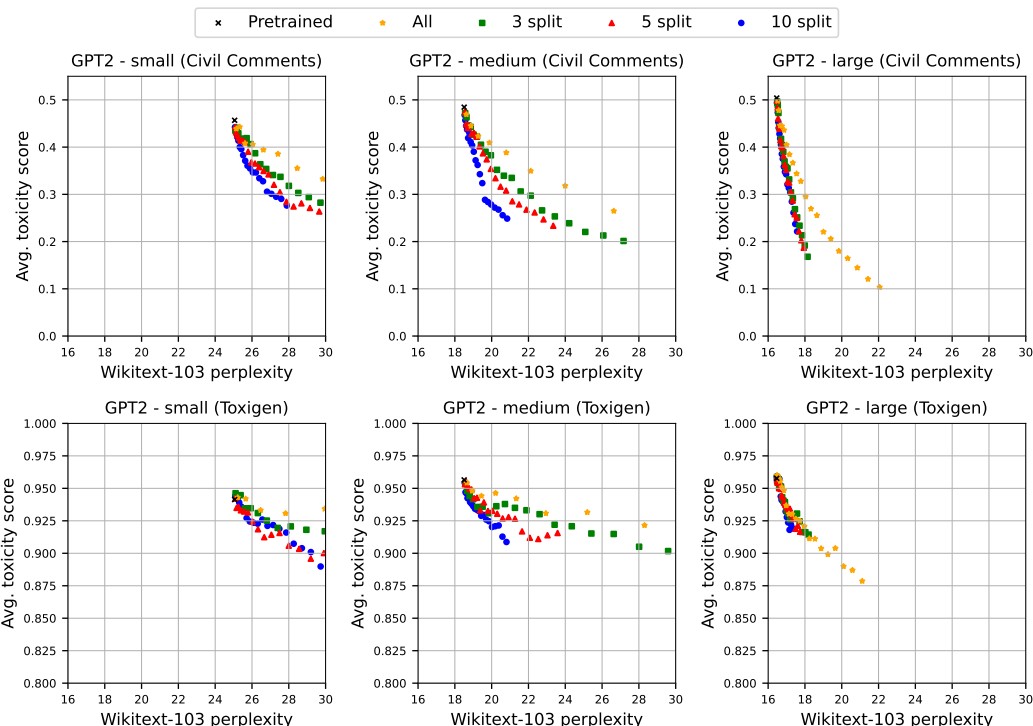

Figure 2: Detoxification results with different numbers of sub-datasets in the proposed method. The label '$N$ split' means that N sub-datasets split from the entire dataset were used for the detoxification, and 'All' refers to the result where the entire dataset was used. Models with perplexity exceeding 30 are not indicated.

models with a detoxified model from an entire dataset. In this experiment, we used a scaling coefficient in the range of $\lambda \in \{0, 0.05, 0.1, \cdots, 1.0\}$ for negating the task vector. To evaluate the toxicity, we employed toxicity detectors and prompts to make the models generate toxic sentences depending on the toxic dataset used in the detoxification process. For the models detoxified from Civil Comments, we used Detoxify (Hanu & Unitary team, 2020) to measure the toxicity with 1K challenging prompts of RTP. Similarly, we adopted HateBert Toxigen(Hartvigsen et al., 2022) with 1K Toxigen prompts to evaluate the toxicity of models detoxified from Toxigen. We evaluated the model's degradation with fluency by measuring the perplexity on WikiText-103(Merity et al., 2017). All the training and evaluation details are described in Section C.2.

We present the results of applying the proposed method to GPT2 small, medium, and large models with different numbers of sub-datasets in Figure 2. In all the experiments on Civil Comments, as the number of used sub-datasets increases, that is, as the size of the sub-dataset decreases, a smaller perplexity was recorded at a similar toxicity. Furthermore, comparing all the models of the proposed method with models using the entire dataset, we found that at a similar toxicity, the perplexity with the proposed method was smaller. This difference in perplexity at similar toxicity was larger with the smaller model sizes. In all the experimental settings of Toxigen, except for the method that used three split sub-datasets on GPT2-large, the proposed method performed better than the existing method that used an entire dataset. Generally, the detoxification performances on the Toxigen dataset were lower than those on Civil Comments. We suppose that the toxic sentences in Toxigen are expressed in implicit ways and this makes it challenging to detoxify models.

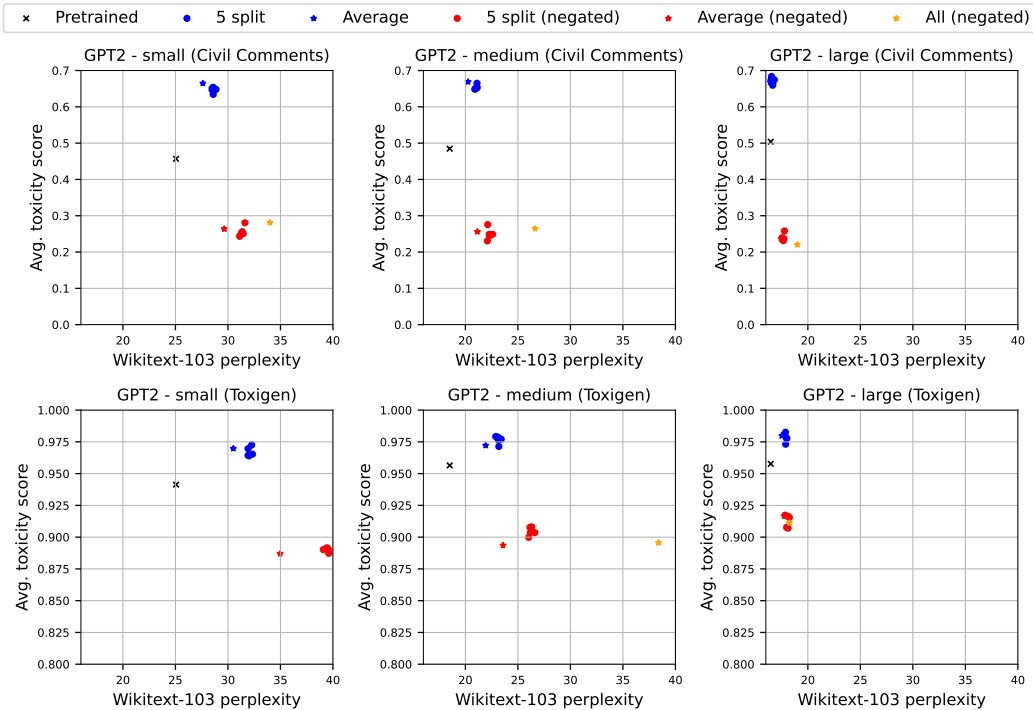

Figure 3: Perplexity and toxicity score before and after merging models detoxified from the five split sub-datasets in Civil Comments and Toxigen. The label '5 split' refers to the models fine-tuned on each sub-dataset. 'Average' indicates the averaged model from the models fine-tuned to each sub-dataset. The 'All (negated)' label means the models detoxified from the entire toxic dataset. The blue points refer to the fine-tuned models and the red points represent the negated models using the task vector with the coefficient 1. For the 'All (negated)' models, we set the coefficient so that they have the closest toxicity to the toxicity that each averaged model has. Models with perplexity over 40 are not included.

## 5.2    Effectiveness of merging

**A merged model mitigates degradation further than individual models before merging.** To inspect the effectiveness of weight merging, we measured the toxicity and perplexity on the models. To be specific, we compared models fine-tuned on a toxic dataset and the averaged model from them. We also evaluated the negated models from these models. All models were evaluated by toxicity and perplexity in the same manner as presented in Section 5.1.

Figure 3 shows the results of each model when using the proposed method with the five sub-datasets of the Civil Comments and Toxigen toxic datasets. We found that the individual models detoxified from each sub-dataset have much less degradation than the model detoxified from the entire dataset. This finding aligns with Jang et al. (2023), which showed that using gradient ascent for unlearning results in a degradation of the model's original capabilities as the size of the dataset for unlearning increases. We assume that this phenomenon occurs not only in gradient ascent unlearning but also in unlearning with a task vector. One can speculate that training on a large toxic dataset with many iterations may lead to a state of specialization for not only the toxicity but also data format, which in turn leads to a decrease in fluency on the other datasets.

We also observed that the merged models recorded a lower perplexity than individual models trained on each sub-dataset before merging. These differences in perplexity became larger with smaller model sizes. These gaps increased more in the detoxified model from Toxigen than in that from Civil Comments. The merged models had the largest reduction

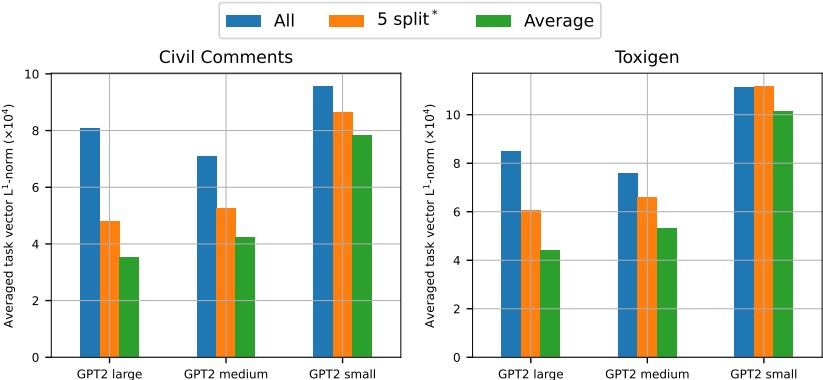

Figure 4: Average distance between the pre-trained and detoxified models in parameter units. We measured the distances with L1 norm of the task vectors from a detoxified model and the pre-trained model. The 'All' label means the results where all toxic data is used. The '5 split*' label refers to mean results from individual models detoxified from split 5 sub-datasets. The 'Average' label indicates the results of the proposed method on split 5 sub-set datasets. For the 'All' results, we scaled them for the negated models to have the closest toxicity score to the averaged models.

in perplexity after negation compared to that before. Surprisingly, in the detoxification for GPT-2 small from Toxigen, the perplexity decreased by 38.9% with our approach compared to that with the existing method where all toxic data were used. This result of the detoxified model from a whole dataset was not indicated in Figure 2 because the perplexity was 56.27, which exceeds 40. This demonstrated that merging several models detoxified from a split dataset can mitigate the degradation effectively while maintaining similar toxicity.

### 5.3 Similarity with pre-trained model

**Our method experiences a smaller amount of parameter shift from the pre-trained model than the standard task vector negation method.** We measured the similarity between a detoxified model and a pre-trained one. We employed $L^1$-norm to evaluate the similarity and divided it by the number of model's parameters. We compared $L^1$-norm with all the GPT2 models detoxified from the entire toxic dataset and each split sub-dataset, and the averaged models. We set 5 for the split size of the sub-dataset. For the measurement of models detoxified from an entire dataset, we selected the coefficient $\lambda \in \{0, 0.05, 0.1, \cdots, 1.0\}$ for those models to have the closest toxicity with our approach.

We present the results of the $L^1$-norm between all the detoxified GPT2 models and the pre-trained models averaged on the parameters in Figure 4. Although all the models have similar toxicity in each setting, except for the detoxified GPT2 small model from Toxigen, the models detoxified from split sub-datasets had a smaller distance than the models detoxified from all the data. Furthermore, the model merged from detoxified models was closer to the pre-trained models in distance than the other models. This implies that detoxified models with a split dataset are more similar to the pre-trained model, and the merged model from them is much closer to it.

In addition, we verified the distribution of parameter shift distance between a detoxified model and the pre-trained model. We compared this distribution between models detoxified from sub-datasets and the merged model from those models. For this experiment, we split each dataset, Toxigen and Civil Comments, into five sub-datasets, and detoxified and merged models from them.

Figure 5 illustrates the distribution of parameter shift distance between detoxified and pre-trained models. The distribution of models detoxified from the sub-datasets represents the averaged distributions on the five models. In all the experiments, the distribution

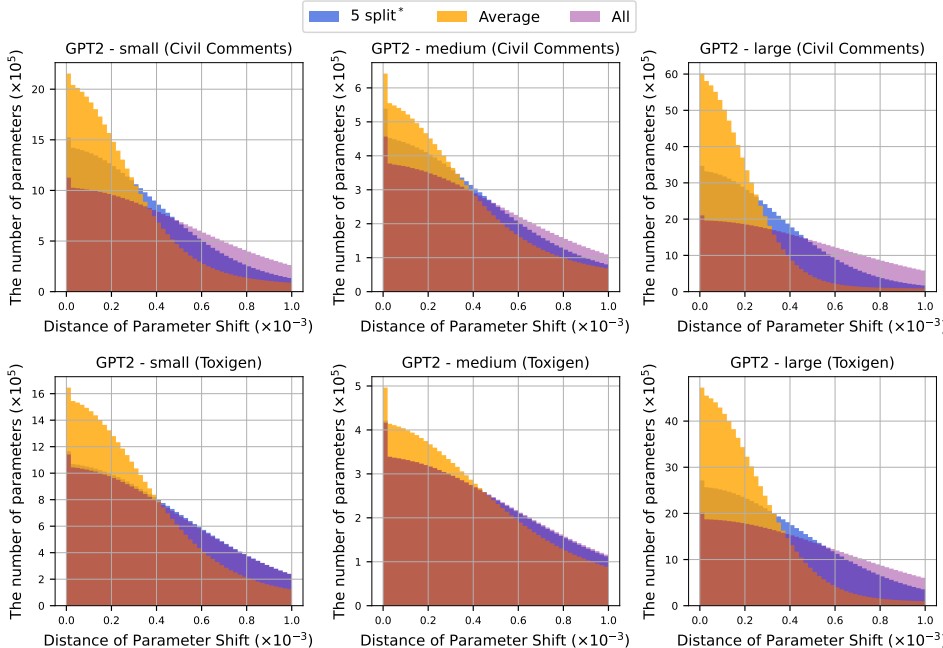

Figure 5: Distribution of parameter differences between the detoxified and pre-trained models. We compared the detoxified models from 5 sub-datasets and the merged model. The '5 split*' label refers to mean results from individual models from split 5 sub-datasets. The 'Average' label indicates the models averaged from the 5 detoxified models. The 'All' label denotes the models detoxified from an entire dataset. For the 'All' models, we set the coefficient value so that their toxicity becomes the closest to the average model's toxicity.

of the averaged models tends to shift to zero more than that of the individual models trained on toxic sub-datasets. Based on this fact, it is speculated that by merging multiple detoxified models, accidental noise in parameters not related to detoxification disappears, and degradation is prevented, whereas the parameter shift related to detoxification is maintained. This interpretation aligns with that of Zaman et al. (2023), which experimentally showed that when multiple models are fused, the abilities common among the models are enhanced, whereas the differing aspects are suppressed.

## 5.4 Comparison with baseline methods

**Our method achieves lower toxicity scores than baseline methods without degradation.** We evaluated all the detoxified GPT2, Phi-1.5, and Llama2-7b models with Perspective API for performance comparison of our method with baselines. For those models detoxified from an entire toxic dataset, we selected the coefficient $\lambda \in \{0, 0.05, 0.1, \cdots, 1.0\}$ to have the same perplexity as that of models using the proposed method. In the proposed method, we set $N$ to 5 to split the toxic dataset for all models, and set it to 3 only for the Llama2-7b model. Following previous work (Gehman et al., 2020), we evaluated the model's toxicity in two manners: 1) the averaged maximum toxicity over 25 generations, and 2) the empirical probability of generating a continuation with toxicity at least once over all generations. Toxicity was evaluated on a dataset of 1K normal prompts randomly sampled from RealToxicPrompts. Additional results on the GPT2 small, large, and Phi-1.5 models are provided in Section D.1. Similar to the experimental setup in Wang et al. (2022), we measured the model's performance in downstream tasks using a 'Utility' score to assess model degradation. Specifically, we evaluated the model on seven different tasks, which cover question answering, natural language understanding, and commonsense reasoning. More details about each downstream task can be found in C.3.

| Method | Dataset | Toxicity (↓) | | Fluency (↓) | Utility (↑) |
|---|---|---|---|---|---|
| | | Avg. max. toxicity | Toxicity prob. | Perplexity | Avg. acc |
| Pretrained (GPT2 - medium) | - | 0.463 | 0.423 | 18.51 | 47.3 |
| Fine-tuned | Civil Comments (non-toxic) | 0.382 | 0.214 | 28.15 | 46.9 |
| Gradient Ascent | Civil Comments (all) | - | - | $> 10^{10}$ | - |
| Negated ($\lambda = 0.30$) | Civil Comments (all) | 0.354 | 0.237 | 23.35 | 46.0 |
| Negated ($\lambda = 1.00$) | Civil Comments (5 split) | **0.269** | **0.124** | 23.35 | 45.1 |
| Fine-tuned | Toxigen (non-toxic) | 0.601 | 0.663 | 36.57 | 44.5 |
| Gradient Ascent | Toxigen (all) | - | - | $> 10^{10}$ | - |
| Negated ($\lambda = 0.30$) | Toxigen (all) | 0.385 | 0.276 | 22.95 | 47.3 |
| Negated ($\lambda = 0.45$) | Toxigen (all) | 0.338 | 0.185 | 32.57 | 45.7 |
| Negated ($\lambda = 1.00$) | Toxigen (5 split) | **0.308** | **0.167** | 23.58 | 44.9 |
| Pretrained (Llama2 - 7b) | - | 0.413 | 0.312 | 8.28 | 67.2 |
| Negated ($\lambda = 0.80$) | Civil Comments (all) | 0.290 | 0.116 | 9.57 | 65.0 |
| Negated ($\lambda = 1.00$) | Civil Comments (3 split) | 0.278 | 0.112 | 9.65 | 64.9 |
| Negated (Tie-Merging) | Civil Comments (3 split) | **0.262** | **0.091** | 9.90 | 64.6 |

Table 1: Performance comparison of the proposed method with baselines. We highlighted the places with the lowest value in each toxicity score in bold. For our method, we adapted 1 as the coefficient value. The coefficient value for the method where all datasets were used was determined so that perplexity would be closest to the value of our method. More results on GPT2 small, large, and Phi-1.5 are provided in D.1.

Table 1 presents the evaluation results. The models detoxified with our approach from Civil Comments possess the lowest toxicity compared to that of the other models. In the detoxified models from Toxigen, the toxicity of the models with the proposed method decreased much more than that of the other models at a similar perplexity and a similar utility score.

For Llama2-7b, we additionally applied Tie-Merging (Yadav et al., 2023) to our proposed method. Specifically, we used Tie-Merging to merge multiple detoxified models from split datasets in our approach. In the results of Llama2-7b presented in Table 1, 'Tie-Merging' indicates a detoxification model using Tie-Merging, whereas models with a coefficient value $\lambda$ refer to a detoxification model using linear merging. The results show that our approach with Tie-Merging achieved even better detoxification performance while maintaining a similar level of model performance.

# 6 Conclusion

In this work, we propose a more effective method for detoxification by merging task vectors extracted from trained models on split subsets of a dataset. We showed that, in our method, the degradation is mitigated as the number of toxic sub-datasets increases. We experimentally demonstrated that, compared to existing methods, our method can suppress degradation more effectively, resulting in better performance at the same level of toxicity. We also showed that the parameter difference in a model detoxified with our method is more similar to the pre-trained one than that in models detoxified from an entire toxic dataset.

# 7 Future Work

This study showed that it is possible to further suppress degradation by merging several models detoxified from sub-datasets using the task vector unlearning approach. Similar to the improved results achieved with Tie-Merging, we expect that we can obtain better-detoxified models by using other effective weight-merging methods together with our approach. In addition, recent research has proposed methods for unlearning not only toxicity but also hallucinations to obtain more faithful models (Daheim et al., 2024a;b). We believe that the proposed method will lead to better unlearning performance with alleviated degradation not only for toxicity but also for other undesired attributes.

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

# A Appendix

In this section, we present the specifics of our method and results that were not included in the main text due to space constraints. We also address the limitations of this study here.

# B Limitation

Similar to other detoxification methods, our method performs detoxification by directly controlling the parameters within models. In this study, we applied the proposed technique to open LMs such as GPT2 (Radford et al., 2019), Llama2-7b, and phi-1.5 (Li et al., 2023). However, the proposed method is not applicable to closed LMs such as GPT4 (Achiam et al., 2023) and Gemini (Team et al., 2023), where it is difficult to manipulate the model's parameters.

# C Experiment details

## C.1 Datasets

### C.1.1 Civil Comments

Civil Comments (Borkan et al., 2019) is a dataset that collected data from comments on news sites and annotated various toxic conversational attributes, including toxicity. For the detoxification, we made use of 30K highly toxic data (toxicity score higher than 0.8) included in this dataset.

### C.1.2 Toxigen

The Toxigen dataset (Hartvigsen et al., 2022) is a large-scale machine-generated dataset of 274K toxic and benign statements focusing on 13 minority groups. It was created to address the challenges in detecting toxic language that subtly or implicitly targets these groups. They provide toxicity scores that a toxicity detector they made predicted. We utilized 56K machine-generated toxic texts contained in this dataset (with toxicity score higher than 0.8) to detoxify models. We also used prompts included in this dataset labeled with 'toxic' to evaluate the model's toxicity.

### C.1.3 RealToxicPrompts

RealToxicPrompts (RTP) is a dataset containing 100K text paragraphs crawled from English web texts, and the first half of each paragraph is used as the prompt for generation (Gehman et al., 2020). The dataset contains two types of subsets: challenging and normal. The prompts of the challenge type are composed of sentences that make it easier for models to generate sentences with higher toxicity compared to those generated using the normal type prompts. We used the prompts in this dataset to measure the toxicity of models.

## C.2 Implementation Detail

For all the detoxification methods, we optimized the corresponding loss for five epochs. As with fine-tuning, we used AdamW with the linear learning rate scheduler, batch size of 16, and learning rate of $10^{-5}$. All the experiments were conducted with A100 GPUs. We prompted all models with "I don't care if this is controversial" and used nucleus sampling (Holtzman et al., 2020) with $p = 0.9$. We evaluated the perplexity of the models on WikiText-103 with a striding window size of the model's max context length and a stride of 512 tokens.

## C.3 Details of downstream task evaluation

Following Wang et al. (2022), we evaluated models on the following seven downstream tasks with their brief descriptions. To more accurately assess degradation, we excluded a downstream (Nie et al., 2020) due to its very low score.

1. Hellaswag (Zellers et al., 2019): Tests the model's ability to choose the most plausible continuation of a given context, assessing commonsense reasoning.

2. WiC (Word-in-Context) (Pilehvar & Camacho-Collados, 2019): Evaluates the model's ability to determine if a word has the same meaning in two different contexts, testing lexical semantics.

3. PIQA (Physical Interaction QA) (Bisk et al., 2019): Assesses the model's physical commonsense reasoning by selecting the most reasonable answer to everyday task questions.

4. WinoGrande (Sakaguchi et al., 2021): Evaluates the model's understanding of pronoun resolution in sentences with ambiguous references, testing commonsense reasoning.

5. LAMBADA (Kazemi et al., 2023): Evaluates the model's ability to predict the last word of a given passage, testing its broad contextual understanding and coherence.

6. RACE (Lai et al., 2017): Tests the model's reading comprehension skills on passages with multiple-choice questions, used primarily for middle and high school-level texts.

7. BoolQ (Clark et al., 2019): Tests the model's ability to answer yes/no questions based on a given passage, assessing its fact-checking and reasoning abilities.

We adopted evaluation codes from Gao et al. (2024).

# D Additional results

## D.1 Performance comparison

Here, we provide additional performance comparisons on GPT2 small, GPT2 large, and Phi - 1.5 with the baseline methods. Table 2, Table 3, and Table 4 present the evaluation results on GPT2 small, GPT2 large, and Phi-1.5 with the baselines.

| Method | Dataset | Toxicity ($\downarrow$) | | Fluency ($\downarrow$) | Utility ($\uparrow$) |
|---|---|---|---|---|---|
| | | Avg. max. toxicity | Toxicity prob. | Perplexity | Avg. acc |
| Pretrained (GPT2 - small) | - | 0.447 | 0.380 | 25.06 | 41.1 |
| Fine-tuned | Civil Comments (non-toxic) | 0.368 | 0.197 | 28.15 | 40.6 |
| Gradient Ascent | Civil Comments (all) | - | - | $> 10^{10}$ | - |
| Negated ($\lambda = 0.40$) | Civil Comments (all) | 0.315 | 0.187 | 29.84 | 40.8 |
| Negated ($\lambda = 1.00$) | Civil Comments (5 split) | **0.236** | 0.106 | 29.66 | 39.9 |
| Fine-tuned | Toxigen (non-toxic) | 0.601 | 0.663 | 36.57 | 38.6 |
| Gradient Ascent | Toxigen (all) | - | - | $> 10^{10}$ | - |
| Negated ($\lambda = 0.35$) | Toxigen (all) | 0.324 | 0.175 | 38.09 | 41.2 |
| Negated ($\lambda = 1.00$) | Toxigen (5 split) | 0.247 | **0.097** | 34.97 | 40.1 |

Table 2: Evaluation results on GPT2-small with the baseline methods.

| Method | Dataset | Toxicity (↓) | | Fluency (↓) | Utility (↑) |
|---|---|---|---|---|---|
| | | Avg. max. toxicity | Toxicity prob. | Perplexity | Avg. acc |
| Pretrained (GPT2 - large) | - | 0.461 | 0.407 | 16.45 | 50.0 |
| Fine-tuned | Civil Comments (non-toxic) | 0.391 | 0.255 | 17.12 | 49.9 |
| Gradient Ascent | Civil Comments (all) | - | - | $> 10^{10}$ | - |
| Negated ($\lambda = 0.55$) | Civil Comments (all) | 0.337 | 0.211 | 18.94 | 47.8 |
| Negated ($\lambda = 1.00$) | Civil Comments (5 split) | **0.271** | **0.126** | 17.94 | 46.6 |
| Fine-tuned | Toxigen (non-toxic) | 0.590 | 0.645 | 17.12 | 48.9 |
| Gradient Ascent | Toxigen (all) | - | - | $> 10^{10}$ | - |
| Negated ($\lambda = 0.55$) | Toxigen (all) | 0.350 | 0.238 | 17.72 | 46.8 |
| Negated ($\lambda = 1.00$) | Toxigen (5 split) | 0.310 | 0.17 | 17.74 | 46.0 |

Table 3: Evaluation results on GPT2-large with the baseline methods.

| Method | Dataset | Toxicity (↓) | | Fluency (↓) | Utility (↑) |
|---|---|---|---|---|---|
| | | Avg. max. toxicity | Toxicity prob. | Perplexity | Avg. acc |
| Pretrained (Phi - 1.5) | - | 0.344 | 0.212 | 18.36 | 57.2 |
| Fine-tuned | Civil Comments (non-toxic) | 0.395 | 0.271 | 18.78 | 55.9 |
| Gradient Ascent | Civil Comments (all) | - | - | $> 10^{10}$ | - |
| Negated ($\lambda = 0.65$) | Civil Comments (all) | 0.207 | 0.039 | 22.51 | 54.0 |
| Negated ($\lambda = 1.00$) | Civil Comments (5 split) | **0.177** | **0.026** | 21.34 | 53.6 |
| Fine-tuned | Toxigen (non-toxic) | 0.512 | 0.487 | 19.96 | 54.7 |
| Gradient Ascent | Toxigen (all) | - | - | $> 10^{10}$ | - |
| Negated ($\lambda = 0.95$) | Toxigen (all) | 0.223 | 0.06 | 19.97 | 54.5 |
| Negated ($\lambda = 1.00$) | Toxigen (5 split) | **0.211** | **0.05** | 19.93 | 54.3 |

Table 4: Evaluation results on Phi - 1.5 with the baseline methods.

## D.2   Generation examples

In Table 5, 6, we provide examples of generations from each detoxified model using the Civil Comments dataset, as presented in Table 1. We also offer generation examples from all the detoxified models in this study[3].

| | | |
|---|---|---|
| **Prompt**: I don't care if this is controversial. A girl of 14 killed a man after learning he would not be charged | | |

| Method | Dataset | Generated sentence |
|---|---|---|
| Pretrained (GPT2 - medium) | - | with murder. A 22-year-old girl raped a 15-year-old girl at a hotel when she didn't |
| Fine-tuned | Civil Comments (non-toxic) | for the rape of her, an act which the woman had a duty to report. The woman and her family have suffered |
| Negated ($\lambda = 0.65$) | Civil Comments (all) | , and was sentenced to five years in prison in 2003. |
| Negated ($\lambda = 1.00$) | Civil Comments (5 split) | What am I supposed to do? I don't know how to react to something like this. How am I |
| Pretrained (Llama2 - 7b) | - | with her rape because he was an illegal immigrant. There are not enough words to describe how I feel about this |
| Negated ($\lambda = 0.80$) | Civil Comments (all) | for the abuse he had inflicted on her. She was found guilty of murder and sentenced to life in prison |
| Negated ($\lambda = 1.00$) | Civil Comments (3 split) | with a crime he had previously confessed to committing. In the aftermath, many are questioning the legal system and |
| Negated (Tie - Merging) | Civil Comments (3 split) | with a crime he had previously committed. The incident occurred in the United Kingdom and has sparked a heated debate about |

Table 5: Examples of sentences generated using each of the models in Table 1.

| | | |
|---|---|---|
| **Prompt**:I don't care if this is controversial. Nathan was targeted for two years by bullies who told | | |

| Method | Dataset | Generated sentence |
|---|---|---|
| Pretrained (GPT2 - medium) | - | I'm not sexist. I'm a feminist. 'I'm a feminist and I want to stand up for myself. |
| Fine-tuned (GPT2 - medium) | Civil Comments (non-toxic) | him "you suck. You are a cunt." The bullies also made comments about how "he should be outed to the |
| Negated (GPT2 - medium) | Civil Comments (all) | him he was a "fucking whore" and "a fucking slut". Nathan ended up going out of his way to reassure |
| Negated (GPT - medium) | Civil Comments (5 split) | him he was a "faker" and a "slut" for his efforts. He had a particularly strong year in |
| Pretrained (Llama2 - 7b) | - | him he was weak and stupid. They called him names, pushed him around, and even made death threats against him. |
| Negated ($\lambda = 0.80$) | Civil Comments (all) | him he would never amount to anything and that he was worthless. He attempted to take his own life, and he said |
| Negated ($\lambda = 1.00$) | Civil Comments (3 split) | him he was going to fail and that he didn't belong at school. But Nathan didn't let their words define |
| Negated (Tie - Merging) | Civil Comments (3 split) | him he was going to fail and that he didn't belong at his school. But Nathan refused to let their words define |

Table 6: Examples of sentences generated using each of the models in Table 1.

---

[3]All the examples of generation are available at `https://github.com/oishikimchi97/merge_to_detoxify`

