# OpenReview forum: "Decoupling Noise and Toxic Parameters for Language Model Detoxification by Task Vector Merging"
_colmweb.org/COLM/2024/Conference — COLM_

### Official Review · Reviewer_VAs3 · 2024-05-05

**Rating:** 7
**Confidence:** 4
**Ethics Flag:** 2

**Summary:**

This paper presents a new approach to detoxify LMs by enhancing task vector merging techniques. The authors propose a method where multiple task vectors, derived from models fine-tuned on separate toxic datasets, are merged. This approach appears to reduce model degradation typically associated with detoxification while maintaining or even improving the LM's ability to suppress toxic output. For instance, using the GPT-2 small model on the Toxigen dataset, the approach achieved a significant reduction in degradation (38.9% less than existing methods) while maintaining comparable levels of toxicity. This suggests that merging multiple detoxified models helps in "decoupling noise and toxic parameters".

**Ethics Concerns Details:**

-	The use of segmented datasets and merged model parameters to detoxify language models raises questions about transparency in how decisions are made within the model. It's important to ensure that these modifications do not obscure the reasoning behind how outputs are generated. -	The possibility of ‘over-censorship’ (false positives) or the suppression of certain forms of expression should be touched upon, dispassionately.

**Questions To Authors:**

-	Can your method scale efficiently with larger models such as GPT-3 or GPT-4, and what computational resources are required for such implementations?
-	Is there a significant increase in training or merging time compared to single-model detoxification approaches?
-	What alternative metrics for model degradation may be used, that were not explored here?

**Reasons To Accept:**

-	This approach is somewhat novel, although combining disparate representations, including those from models trained on different-but-related tasks is not really new, broadly
-	There is good empirical evidence provide to demonstrate the utility of the model. In particular, it is commendable that so many of these models are open-source and accessible to a broader researcher community than those that rely on closed, corporate models
-	There are potential applications beyond model detoxification, specifically. As noted in the paper, this approach may generalize to other ML tasks where some metric performance must be maintained but other behaviours mitigated against. This is nevertheless the subject of future work.

**Reasons To Reject:**

-	The method's efficacy is demonstrated primarily through reductions in toxicity scores and model degradation rates, but there's a lack of discussion on whether the segmentation of datasets and subsequent model merging might lead to overfitting on the specific characteristics of the datasets used (i.e., Civil Comments and Toxigen). What other aspects of these datasets are being exploited in this process? Without further clarity, this could impact the robustness of the method when applied to real-world, diverse datasets (Section 4.1).
-	Despite the empirical rigour in general, this paper does not explore a wider collection of datasets (or languages) nor does it provide a robust collection of baselines (i.e., existing detoxification methods, especially newer or more complex architectures) against which results are compared.

---

> ### Author Rebuttal · Authors · 2024-05-31
>
> > Can your method scale efficiently with larger models such as GPT-3 or GPT-4, and what computational resources are required for such implementations?
>
> It’s hard to adapt closed source LLMs such as GPT-3, GPT4 because our approach needs to merge parameters from several models, which is not available in closed source LLMs. However, We believe our method can be applied to open-sourced larger models efficiently. Our approach requires the same amount of computation as the existing task vector negation method, except for merging multiple detoxified models, because total number of iterations of training multiple models on the split datasets is adjusted to be the same as the number of iterations for training a single model on the entire dataset is  the same as total number of iterations of training multiple models on the split datasets. The process of merging multiple models requires significantly less computation than the training process and can be performed using only a single CPU. Moreover, this can be done more effectively by using off-the-shelf libraries such as Mergekit.
>
> > Is there a significant increase in training or merging time compared to single-model detoxification approaches?
>
> As we mentioned in our previous answer, the training time is the same compared to the single-model detoxification method and our approach only requires additional time for the merging process. However, we believe that the merging time is relatively minor compared to the training time. Moreover, using the off-the-shelf libraries for merging models can make this process even more efficient.
>
> > Paper lacks comparison to other strong model self-detoxification approaches...
>
> We did not compare our method with strong self-detoxification approaches as mentioned by the reviewer. Those methods require creating extensive toxic and non-toxic datasets for each model, which is data-intensive. Our task vector negation method allows efficient detoxification using only existing toxic datasets. We measured the performance of a typical detoxification method using only toxic datasets and confirmed our method’s superiority under the assumption of the same fine-tuning data availability.
>
> Due to length limitations, we were unable to answer all of the reviewer’s questions and cannot include references.
> We appreciate the reviewer’s understanding and suggest referring to our responses to the other reviewers

---

> > ### Comment · Reviewer_VAs3 · 2024-06-05
> >
> > Thank you -- these are clear responses to my questions

---

> ### Author Response · Authors · 2024-05-31
> **Additional Response to Reviewer VAs3**
>
> Dear Reviewer VAs3,
>
> We sincerely apologize that we could not provide an enough response on the Rebuttal due to the length limitation.
> In this comment, we will address the parts that were lacking or unanswered in the Rebuttal. We hope this will help for Reviewers.
>
> We wish to express our appreciation to the reviewers for their insightful comments on our paper. Please find our responses below.
>
> > The method's efficacy is demonstrated primarily through reductions in toxicity scores and model degradation rates, but there's a lack of discussion on whether the segmentation of datasets and subsequent model merging might lead to overfitting on the specific characteristics of the datasets used (i.e., Civil Comments and Toxigen). What other aspects of these datasets are being exploited in this process? Without further clarity, this could impact the robustness of the method when applied to real-world, diverse datasets (Section 4.1).
>
> > What alternative metrics for model degradation may be used, that were not explored here?
>
> As the reviewer mentioned, we acknowledge the possibility of overfitting to other characteristics of the dataset beyond toxicity. However, as we mentioned in this paper, our method has been shown to enhance a common feature (detoxification) and suppress distinct features (data format, etc.) in segmented datasets, which we believe will result in better generality compared to the existing task vector negation method. To further clarify the degeneration of other aspects, we have included additional evaluations on various downstream tasks [1, 2, 3, 4, 5, 6, 7]. We choose these downstream tasks for the evaluation of model degeneration following this detoxification research [8]. We measure this by averaging accuracy on the downstream tasks. We provide a short description for each downstream task on [**our comment to Reviewer tRLA**](https://openreview.net/forum?id=TBNYjdOazs&noteId=04JTMxr8VH). Moreover, we have presented the main experiment results including the downstream evaluations in the table (please see [**our Rebuttal to Reviewer tRLA**](https://openreview.net/forum?id=TBNYjdOazs&noteId=dHnDleniE3)). As a result, We found that our approach can achieve better detoxification than the existing task vector negation method while maintaining similar model performance.
>
> > Despite the empirical rigour in general, this paper does not explore a wider collection of datasets (or languages) nor does it provide a robust collection of baselines (i.e., existing detoxification methods, especially newer or more complex architectures) against which results are compared.
>
> Due to the limited number of existing toxic datasets, as the reviewer mentioned, we were not able to conduct experiments with a wider variety of datasets. However, to ensure the robustness of our method across datasets, we included the Toxigen in addition to the Civil Comments dataset used in previous detoxification research. Furthermore, we conducted experiments with Llama2-7b and presented the results in [**our Rebuttal to Reviewer vKrR**](https://openreview.net/forum?id=TBNYjdOazs&noteId=qLvE2RNltY). The results in that table show that the proposed method is also effective on the recent and larger model.
>
> Regarding the comparison with existing detoxification methods, we did not compare our proposed method with existing strong self-detoxification approaches [8, 9]. However, these self-detoxification methods require the creation of extensive toxic and non-toxic datasets from a model. These datasets need to be generated for each model, requiring a large amount of data for each detoxification. However, the task vector negation method adopted in the proposed approach allows for efficient detoxification using only existing toxic datasets.
>
> Thank you for your advice. We will update our paper following your advice.
>
> [1] [HellaSwag: Can a Machine Really Finish Your Sentence?](https://arxiv.org/abs/1905.07830) by Zellers et al.
>
> [2] [WiC: the Word-in-Context Dataset for Evaluating Context-Sensitive Meaning Representations](http://arxiv.org/abs/1808.09121) by Pilehvar and Camacho-Collados
>
> [3] [PIQA: Reasoning about Physical Commonsense in Natural Language](http://arxiv.org/abs/1911.11641) by Bisk et al.
>
> [4] [WinoGrande: An Adversarial Winograd Schema Challenge at Scale](http://arxiv.org/abs/1907.10641) by Sakaguchi et al.
>
> [5] [LAMBADA: Backward Chaining for Automated Reasoning in Natural Language](http://arxiv.org/abs/2212.13894) by Kazemi et al.
>
> [6] [RACE: Large-scale ReAding Comprehension Dataset From Examinations](http://arxiv.org/abs/1704.04683) by Lai et al.
>
> [7] [BoolQ: Exploring the Surprising Difficulty of Natural Yes/No Questions](http://arxiv.org/abs/1905.10044) by Clark et al.
>
> [8] [Exploring the limits of domain-adaptive training for detoxifying large-scale language models](https://arxiv.org/abs/2202.04173) by Wang et al.
>
> [9] [CMD: a framework for Context-aware Model self-Detoxification](http://arxiv.org/abs/2308.08295) by Tang et al.

---

### Official Review · Reviewer_o8PJ · 2024-05-10

**Rating:** 4
**Confidence:** 4
**Ethics Flag:** 1

**Summary:**

The paper proposed a method of toxicity mitigation by merging several models fine-tuned on toxic data. Several models are trained on different subsets of toxic examples and then averaged and subtracted from the initial model. Results show that this technique is able to reduce toxic generations up to 38% without noticeable performance degradation.

**Questions To Authors:**

- Have you tried this method on larger models? Does it work at scale?
- What kind of toxic phenomena is evaluated? Words that are considered toxic or sentences that are toxic due to the context?

**Reasons To Accept:**

- Toxicity mitigation is an important topic now that LLMs are becoming commercial products.
- The proposed method is simple and easy to apply to other models.

**Reasons To Reject:**

- The paper contains several typos ("detoxify the each model", "paramter").
- The models evaluated are relatively small or non state-of-the-art. It would be interesting to see if this technique scales on bigger models.
- The paper does not provide a clear definition of toxicity or the kind of phenomena evaluated. It would be interesting to know some examples of toxicity evaluated and how accurate is Detoxify on those examples.
- Perplexity is not currently considered the best metric for generation performance. The model could produce outputs with high confidence while outputting low-quality text. It would be interesting to measure the performance of the model on some LLM benchmarks to ensure the lack of performance degradation.

---

> ### Author Rebuttal · Authors · 2024-05-31
>
> > The paper contains several typos ("detoxify the each model", "paramter").
>
> We sincerely apologize for any difficulties in following our experiment settings. Based on advice of many reviewers, if our paper is accepted, we plan to make the following changes and additions in the camera-ready version:
>
> 1. Correct typos and grammar errors.
> 2. Clearly describe the purpose and intent of each experiment.
> 3. Provide more detailed information on the evaluation methods and settings for each experiment  (Please also see our response below).
> 4. Include a precise definition of toxicity and examples for each toxicity category  (Please also see our response from Reviewer vKrR).
>
> > The paper does not provide a clear definition of toxicity or the kind of phenomena evaluated. It would be interesting to know some examples of toxicity evaluated and how accurate is Detoxify on those examples.
> > What kind of toxic phenomena is evaluated? Words that are considered toxic or sentences that are toxic due to the context?
>
> We sincerely apologize for any confusion caused by not clearly defining toxicity. The definition of toxicity is subjective and may vary slightly across different studies, but we used the toxicity definition from the Perspective API. According to the Perspective API, which is commonly used to assess toxicity in many papers, toxicity is defined as “a rude, disrespectful, or unreasonable comment that is likely to make you leave a discussion.” The Perspective API categorizes toxicity into four levels and classifies toxic sentences into four categories. The Perspective API uses data labeled by humans according to their toxic levels and categories. This data is used to train a classifier, which then measures the toxicity of text. If our paper is accepted, we plan to include examples of generated sentences from both the baseline and our detoxified models, categorized by the toxicity levels and the categories. Once again, we apologize for not providing a clear definition and promise to include a more precise definition in the paper.
>
> Due to length limitations, we were unable to answer all of the reviewer’s questions and cannot include references.
> We appreciate the reviewer’s understanding and suggest referring to our responses to the other reviewers

---

> ### Author Response · Authors · 2024-05-31
> **Additional Response to Reviewer o8PJ**
>
> Dear Reviewer o8PJ,
>
> We sincerely apologize that we could not provide an enough response on the Rebuttal due to the length limitation.
> In this comment, we will address the parts that were lacking or unanswered in the Rebuttal. We hope this will help for Reviewers.
> We wish to express our appreciation to the reviewers for their insightful comments on our paper. Please find our responses below.
>
> > Perplexity is not currently considered the best metric for generation performance. The model could produce outputs with high confidence while outputting low-quality text. It would be interesting to measure the performance of the model on some LLM benchmarks to ensure the lack of performance degradation.
>
> Following the feedback from the reviewers, we have included additional evaluations on various downstream tasks [1, 2, 3, 4, 5, 6, 7]. We choose these downstream tasks for the evaluation of model degeneration following this detoxification research [8]. We measure this by averaging accuracy on the downstream tasks. We provide a short description of these downstream tasks on [**our comments to Reviewer tRLA**](https://openreview.net/forum?id=TBNYjdOazs&noteId=04JTMxr8VH). Moreover, we have presented the main experiment results including the downstream evaluations on [**our rebuttal to Reviewer tRLA’**](https://openreview.net/forum?id=TBNYjdOazs&noteId=dHnDleniE3). As a result, We found that our approach can achieve better detoxification than the existing task vector negation method, while maintaining similar model performance.
>
> > The models evaluated are relatively small or non state-of-the-art. It would be interesting to see if this technique scales on bigger models.
> > Have you tried this method on larger models? Does it work at scale?
>
> We believe that our approach will also work on larger and state-of-the-art models. To demonstrate this, we applied our proposed method to a larger and more recent model, Llama2 - 7b, and presented the results in the table (please see the table in [**our Rebuttal to Reviewer vKrR**](https://openreview.net/forum?id=TBNYjdOazs&noteId=qLvE2RNltY)). The results in that table show that the proposed method is also effective on the recent and larger model.
>
> Thank you for your advice. We will update our paper following your advice.
>
> Again, thank you for giving us the opportunity to improve our manuscript with your valuable comments and queries.
>
>
> [1] [HellaSwag: Can a Machine Really Finish Your Sentence?](https://doi.org/10.18653/v1/P19-1472) by Zellers et al.
>
> [2] [WiC: the Word-in-Context Dataset for Evaluating Context-Sensitive Meaning Representations](http://arxiv.org/abs/1808.09121) by Pilehvar and Camacho-Collados
>
> [3] [PIQA: Reasoning about Physical Commonsense in Natural Language](http://arxiv.org/abs/1911.11641) by Bisk et al.
>
> [4] [WinoGrande: An Adversarial Winograd Schema Challenge at Scale](http://arxiv.org/abs/1907.10641) by Sakaguchi et al.
>
> [5] [LAMBADA: Backward Chaining for Automated Reasoning in Natural Language](http://arxiv.org/abs/2212.13894) by Kazemi et al.
>
> [6] [RACE: Large-scale ReAding Comprehension Dataset From Examinations](http://arxiv.org/abs/1704.04683) by Lai et al.
>
> [7] [BoolQ: Exploring the Surprising Difficulty of Natural Yes/No Questions](http://arxiv.org/abs/1905.10044) by Clark et al.
>
> [8] [Exploring the limits of domain-adaptive training for detoxifying large-scale language models](https://proceedings.neurips.cc/paper_files/paper/2022/file/e8c20cafe841cba3e31a17488dc9c3f1-Paper-Conference.pdf) by Wang et al.

---

> ### Author Response · Authors · 2024-06-07
> **A Gentle Reminder to o8PJ**
>
> **Dear Reviewer o8PJ,**
>
> As of now, we have not received any responses from the reviewer. If the reviewer has any questions or comments about the rebuttal or the results of the additional experiments, we would be happy to answer them. We apologize for the last-minute reminder during your busy time.

---

### Official Review · Reviewer_tRLA · 2024-05-11

**Rating:** 7
**Confidence:** 4
**Ethics Flag:** 1

**Summary:**

This paper proposes a novel approach combining model merging and task arithmetic-based detoxification. The proposed method has the following steps: a) a toxicity dataset is split, and the toxicity task is learned from the splits by the pre-trained models; b) the learned models are then merged, and the toxicity task vector is learned, and c) the task vector is subtracted from the pre-trained model. The primary claim is that this reduces model degradation while achieving better detoxification.

**Reasons To Accept:**

* The work proposes an interesting new angle to the detoxification of large language models by combining two different lines of research
* The proposed method shows promising results in reducing toxicity compared to other methods, such as fine-tuning and gradient ascent

**Reasons To Reject:**

* The primary issue with the paper is its writing. The exact methodology is unclear and hard to follow. The experimental setup does not clearly state the experimental setup, choice of metrics for evaluation, and justification for choices made in the experimental design.
* Beyond writing, the first point of confusion is how toxicity is evaluated. How is avg. max toxicity and toxicity probs are calculated, as shown in Table 1.
* Degeneration of the model is evaluated with perplexity on wiki-103. This metric alone cannot sufficiently support the claim that this method leads to lesser degeneration. For example, what happens to other capabilities of the models? It would also be important to evaluate the generation themselves, as perplexity could be a very noisy proxy for generation quality. Specifically, human evaluation of generations would be highly recommended.
* I think this work has a really good premise. I encourage the authors to add more experiments on generation quality, polishing the writing, and resubmitting it at a future conference.

---

> ### Author Rebuttal · Authors · 2024-05-31
>
> > Degeneration of the model is evaluated with perplexity on wiki-103. This metric alone cannot sufficiently support the claim that this method leads to lesser degeneration. For example, what happens to other capabilities of the models?...
>
> Following the feedback from the reviewers, we have included additional evaluations on various downstream tasks. We choose these downstream tasks for the evaluation of model degeneration following this detoxification research. We measure this by averaging accuracy on the downstream tasks.
>
> We have presented the main experiment results including the downstream evaluations in the table below.
>
>
> | Method                       |           Dataset          | Toxicity ($\downarrow$) |                | Fluency ($\downarrow$) | Utility ($\uparrow$) |
> |------------------------------|:--------------------------:|:-----------------------:|:--------------:|:----------------------:|:--------------------:|
> |                              |                            |    Avg. max. toxicity   | Toxicity prob. |       Perplexity       |       Avg. acc       |
> | Pretrained (GPT2 - medium)   |              -             |          0.463          |      0.423     |          18.51         |         47.3         |
> | Fine-tuned                   | Civil Comments (non-toxic) |          0.382          |      0.214     |          28.15         |         46.9         |
> | Gradient Ascent              |    Civil Comments (all)    |            -            |        -       |       $>10^{10}$       |           -          |
> | Negated ($\lambda$ = 0.30)   |    Civil Comments (all)    |          0.354          |      0.237     |          23.35         |         46.0         |
> | Negated ($\lambda$ = 1.00)   |  Civil Comments (5 split)  |          0.269          |      0.124     |          23.35         |         45.1         |
>
>
> We indicated the average scores for downstream tasks as the ‘Utility’ score in the rightmost column. Our model’s degeneration after detoxification was similar to the existing task vector negation method but achieved better detoxification. We will include the detailed evaluation results of all detoxified models in the final version if our paper is accepted.
>
> Due to length limitations, we were unable to answer all of the reviewer’s questions and cannot include references.
> We appreciate the reviewer’s understanding and suggest referring to our responses to the other reviewers

---

> ### Author Response · Authors · 2024-05-31
> **Additional Response to Reviewer tRLA**
>
> Dear Reviewer tRLA,
>
> We sincerely apologize that we could not provide an enough response on the Rebuttal due to the length limitation.
> In this comment, we will address the parts that were lacking or unanswered in the Rebuttal. We hope this will help for Reviewers.
> We wish to express our appreciation to the reviewers for their insightful comments on our paper. Please find our responses below.
>
> >The primary issue with the paper is its writing. The exact methodology is unclear and hard to follow. The experimental setup does not clearly state the experimental setup, choice of metrics for evaluation, and justification for choices made in the experimental design.
>
> We sincerely apologize for any difficulties in following our experiment settings. Based on the advice of many reviewers, if our paper is accepted, we plan to make the following changes and additions in the camera-ready version:
>
> 1. Correct typos and grammatical errors.
> 2. Clearly describe the purpose and intent of each experiment.
> 3. Provide more detailed information on the evaluation methods and settings for each experiment (Please see our response below).
> 4. Include a precise definition of toxicity and examples for each toxicity category (Please also see [**our comment to Reviewer vKrR**](https://openreview.net/forum?id=TBNYjdOazs&noteId=X3MKeMdDna)).
>
> Once again, we deeply apologize for the unclear explanations that made it difficult to comprehend this paper. If there are any other comments in addition to what we have mentioned, please let us know. Thank you very much.
>
> >Beyond writing, the first point of confusion is how toxicity is evaluated. How is avg. max toxicity and toxicity probs are calculated, as shown in Table 1.
>
> We are truly sorry for the difficulty caused by not providing clear explanations for the metrics in the table caption. Following the previous study [1], We evaluate the model’s toxicity using Perspective API [2] in Table 1. We specifically use the nucleus sampling strategy [3] to produce 25 generations of 25 tokens each for the same prompt. This allows us to determine the maximum toxicity score for each prompt and compute the average as the Expected Maximum Toxicity. Additionally, we calculate the probability of generating toxic continuations (toxicity score above 0.5) within the 25 generations for each prompt, referred to as the Toxicity Probability score.
>
> We prompt all models with “I don’t care if this is controversial” and we use 1K prompts labeled as ‘normal’, and randomly sampled from RealToxicPrompts [1] for the toxicity evaluation. Although the explanations for these metrics are included in the paper, we have realized that it is challenging for readers to understand without them being under the table caption. We will address this issue with the other writing problems the reviewers have pointed out, in the camera-ready version.
>
> > Degeneration of the model is evaluated with perplexity on wiki-103. This metric alone cannot sufficiently support the claim that this method leads to lesser degeneration. For example, what happens to other capabilities of the models? It would also be important to evaluate the generation themselves, as perplexity could be a very noisy proxy for generation quality. Specifically, human evaluation of generations would be highly recommended.
>
> Although we also agree with the reviewer’s opinion that human evaluation is the most accurate evaluation, we were unable to conduct human evaluation due to time constraints. However, following the feedback from the reviewers, we have included additional evaluations on various downstream tasks [4, 5, 6, 7, 8, 9, 10]. We choose these downstream tasks for the evaluation of model degeneration following this detoxification research [11]. We measure this by averaging accuracy on the downstream tasks. We provide short descriptions for each downstream task in the next comment due to the length limitation.

---

> ### Author Response · Authors · 2024-05-31
> **Additional Response 2 to Reviewer tRLA**
>
> Continuing from the previous comment, we have provided a brief description of the downstream tasks below.
>
> Hellaswag [4]: Tests the model’s ability to choose the most plausible continuation of a given context, assessing commonsense reasoning.
> WiC (Word-in-Context) [5]: Evaluates the model’s ability to determine if a word has the same meaning in two different contexts, testing lexical semantics.
> PIQA (Physical Interaction QA) [6]: Evaluates the model’s physical commonsense reasoning by selecting the most reasonable answer to everyday task questions.
> WinoGrande [7]: Evaluates the model’s understanding of pronoun resolution in sentences with ambiguous references, testing commonsense reasoning.
> LAMBADA [8]: Evaluates the model’s ability to predict the last word of a given passage, testing its broad contextual understanding and coherence.
> RACE [9]: Tests the model’s reading comprehension skills on passages with multiple-choice questions, used primarily for middle and high school-level texts.
> BoolQ [10]: Tests the model’s ability to answer yes/no questions based on a given passage, assessing its fact-checking and reasoning abilities.
>
> In the rebuttal section, we have presented the main experiment results including the downstream evaluations in the table. We truly apologize for not being able to present the results clearly due to the length limitation.
>
> Thank you for your advice. We will update our paper following your advice.
> Again, thank you for giving us the opportunity to improve our manuscript with your valuable comments and queries.
>
> [1] [RealToxicityPrompts: Evaluating Neural Toxic Degeneration in Language Models](https://doi.org/10.18653/v1/2020.findings-emnlp.301) by Gehman et al.
>
> [2] [Perspective API](https://github.com/conversationai/perspectiveapi) by Conversation AI
>
> [3] [The curious case of neural text degeneration](https://openreview.net/forum?id=rygGQyrFvH) by Holtzman et al.
>
> [4] [HellaSwag: Can a Machine Really Finish Your Sentence?](https://aclanthology.org/P19-1472) by Zellers et al.
>
> [5] [WiC: the Word-in-Context Dataset for Evaluating Context-Sensitive Meaning Representations](http://arxiv.org/abs/1808.09121) by Pilehvar and Camacho-Collados
>
> [6] [PIQA: Reasoning about Physical Commonsense in Natural Language](http://arxiv.org/abs/1911.11641) by Bisk et al.
>
> [7] [WinoGrande: An Adversarial Winograd Schema Challenge at Scale](http://arxiv.org/abs/1907.10641) by Sakaguchi et al.
>
> [8] [LAMBADA: Backward Chaining for Automated Reasoning in Natural Language](http://arxiv.org/abs/2212.13894) by Kazemi et al.
>
> [9] [RACE: Large-scale ReAding Comprehension Dataset From Examinations](http://arxiv.org/abs/1704.04683) by Lai et al.
>
> [10] [BoolQ: Exploring the Surprising Difficulty of Natural Yes/No Questions](http://arxiv.org/abs/1905.10044) by Clark et al.

---

> ### Author Response · Authors · 2024-06-07
> **A Gentle Reminder to tRLA**
>
> **Dear Reviewer tRLA,**
>
> As of now, we have not received any responses from the reviewer. If the reviewer has any questions or comments about the rebuttal or the results of the additional experiments, we would be happy to answer them. We apologize for the last-minute reminder during your busy time.

---

> > ### Comment · Reviewer_tRLA · 2024-06-07
> >
> > Thank you for addressing all my concerns and questions. Thanks.

---

### Official Review · Reviewer_vKrR · 2024-05-14

**Rating:** 6
**Confidence:** 4
**Ethics Flag:** 1

**Summary:**

The paper introduces an approach for language model detoxification by fine-tuning multiple instances of the model on the detoxification dataset splits and then merging the subtracted detoxification vectors. Authors test their approach on GPT-2 (small, medium and large) and Phi-1.5.

**Reasons To Accept:**

-  The paper is well-written, though there are several typos and grammatical errors and I encourage authors to refine the text.
- The experiments cover a list of language model types, sizes (in the case of GPT-2) and datasets, demonstrating the generalizability of the approach. Though, it would be more beneficial to consider language models, that are used in other works and add comparison to competing approaches as well.
- The paper provides a thorough analysis of the parameter shift between the detoxified models and the pre-trained models, showing that the proposed method results in smaller deviations.
-  The methodology is clearly explained, making it easy to understand. The code is also provided which will help to other reader to reproduce the results.

**Reasons To Reject:**

- **Novelty and overall contribution.** The idea of using task vectors is not novel and have been widely explored in the literature [1, 2, 3]. Moreover, learning a task vector for a text style transfer task is also not novel [4]. Therefore, the contribution
- **Lack of other merging strategies.** Authors do not consider any other merging strategies except linear. It would be beneficial to consider other merging strategies from related work [5, 6]. These methods were proven to be better than the simplest linear merging. Though pointed out in Future work section, this limitation appears to be crucial for the work to be considered thorough enough.
- **Lack of strong baselines or competing approaches.** Paper lacks comparison to other strong model self-detoxification approaches, e.g. [7]. With the current evaluation results it is unclear, whether the proposed method is better than other strong approaches.


References:
[1] [Editing Models with Task Arithmetic](https://arxiv.org/abs/2212.04089) by Ilharco et al.

[2] [In-Context Learning Creates Task Vectors](https://aclanthology.org/2023.findings-emnlp.624/) by Hendel et al.

[3] [Towards Universality in Multilingual Text Rewriting](https://arxiv.org/abs/2107.14749) by Garcia et al.

[4] [Specializing Small Language Models towards Complex Style Transfer via Latent Attribute Pre-Training](https://arxiv.org/abs/2309.10929) by Xu et al.

[5] [Merging Models with Fisher-Weighted Averaging](https://arxiv.org/abs/2111.09832) by Matena et al.

[6] [Dataless Knowledge Fusion by Merging Weights of Language Models](https://arxiv.org/abs/2212.09849) by Jin et al.

[7] [CMD: a framework for Context-aware Model self-Detoxification](https://arxiv.org/abs/2308.08295) by Tang et al.

---

> ### Author Rebuttal · Authors · 2024-05-31
>
> > Authors do not consider any other merging strategies except linear. It would be beneficial to consider other merging strategies from related work. These methods were proven to be better than the simplest linear merging. Though pointed out in Future work section, this limitation appears to be crucial for the work to be considered thorough enough.
>
> Following the opinions of the reviewer, we applied Ties-Merging to our proposed method, which is widely used as a merging method other than linear merging. Specifically, we used Tie-Merging to merge multiple detoxified models from split datasets in our approach. We have presented the results including this detoxification method to Llama2-7b in the table below.
>
>
>
> | Method                             |         Dataset         | Toxicity ($\downarrow$) |                | Fluency ($\downarrow$) | Utility ($\uparrow$) |
> |------------------------------------|:------------------------:|:-----------------------:|:--------------:|:----------------------:|:--------------------:|
> |                                    |                          |    Avg. max. toxicity   | Toxicity prob. |       Perplexity       |       Avg. acc       |
> | Pretrained (Llama2 - 7b)           |             -            |          0.413          |      0.312     |          8.28          |         67.2         |
> | Negated (Linear, $\lambda$ = 0.80) |   Civil Comments (all)   |          0.290          |      0.116     |          9.57          |         65.0         |
> | Negated (Linear, $\lambda$ = 1.00) | Civil Comments (3 split) |          0.278          |      0.112     |          9.65          |         64.9         |
> | Negated (Ties - Merging)           | Civil Comments (3 split) |          0.262          |      0.091     |          9.90          |         64.6         |
>
>
>
> In the table, “Linear” refers to linear merging, and Ties-Merging represents a detoxification model using Tie -Merging. Following the advice from the reviewer, we have also included the utility score, which indicates the average scores on downstream tasks. The results in the table show that our approach with Ties-Merging also achieved better detoxification performance while maintaining a similar level of model performance (Fluency, Utility scores).
>
> Due to length limitations, we were unable to answer all of the reviewer’s questions and cannot include references.
> We appreciate the reviewer’s understanding and suggest referring to our responses to the other reviewers

---

> ### Author Response · Authors · 2024-05-31
> **Additional Response to Reviewer vKrR**
>
> Dear Reviewer vKrR,
>
> We sincerely apologize that we could not provide an enough response on the Rebuttal due to the length limitation.
> In this comment, we will address the parts that were lacking or unanswered in the Rebuttal. We hope this will help for Reviewers.
>
> We wish to express our appreciation to the reviewers for their insightful comments on our paper. Please find our responses below.
>
> > The idea of using task vectors is not novel and have been widely explored in the literature. Moreover, learning a task vector for a text style transfer task is also not novel. Therefore, the contribution
>
> As the reviewer has pointed out, we conducted detoxification by utilizing task vector negation. However, we believe that our main contributions are as follows. 1) We proposed a novel detoxification method by merging multiple task vectors from fine-tuned models with split toxic datasets, and 2) We discovered that when models trained on split datasets are merged, the merged model experiences a smaller amount of parameter shift from the pre-trained model than the standard task vector negation method. We assume that by applying our method, the important attribute (detoxification) is maintained within the merged vectors, while other attributes (text format, etc.), which we believe might be present in the split datasets and related to model degeneration, are suppressed.
>
> > Lack of other merging strategies. Authors do not consider any other merging strategies except linear. It would be beneficial to consider other merging strategies from related work. These methods were proven to be better than the simplest linear merging.
>
> The detoxification results are shown in the table in the Rebuttal section. We mentioned the performance (Utility Score) on downstream tasks and described these tasks in [**our comments to Reviewer tRLA**](https://openreview.net/forum?id=TBNYjdOazs&noteId=04JTMxr8VH). We hope this will be helpful.
>
> > The paper does not provide a clear definition of toxicity or the kind of phenomena evaluated. It would be interesting to know some examples of toxicity evaluated and how accurate is Detoxify on those examples.
>
> We sincerely apologize for any confusion caused by not clearly defining toxicity. The definition of toxicity is subjective and may vary slightly across different studies, but we used the toxicity definition from the Perspective API [2]. According to the Perspective API [2], which is widely used to assess toxicity in many papers [1, 3, 4], toxicity is defined as “a rude, disrespectful, or unreasonable comment that is likely to make you leave a discussion.” The Perspective API categorizes toxicity into four levels and classifies toxic sentences into four categories. The Perspective API uses data labeled by humans according to their toxic levels and categories. This data is used to train a classifier, which then measures the toxicity of text. If our paper is accepted, we plan to include examples of generated sentences from both the baseline and our detoxified models, categorized by the toxicity levels and the categories. Once again, we apologize for not providing a clear definition and promise to include a more precise definition in the paper.
>
> > Paper lacks comparison to other strong model self-detoxification approaches. With the current evaluation results it is unclear, whether the proposed method is better than other strong approaches.
>
> In this paper, we did not compare our proposed method with existing strong self-detoxification approaches [3, 4], as mentioned by the reviewer. However, these self-detoxification methods require the creation of both extensive toxic and non-toxic datasets from a model. These datasets need to be generated for each model, requiring a large amount of data for each detoxification. However, the task vector negation method adopted in our proposed approach allows for efficient detoxification using only existing toxic datasets. We measured the performance of a typical detoxification method (i.e., gradient ascent) using only toxic datasets and compared its performance with our proposed method, confirming that our method is superior to those under the assumption that the fine-tuning data available are the same.
>
>
> [1] [Editing Models with Task Arithmetic](http://arxiv.org/abs/2212.04089) by Ilharco et al.
>
> [2] [Perspective API](https://github.com/conversationai/perspectiveapi) by Conversation AI
>
> [3] [CMD: a framework for Context-aware Model self-Detoxification](http://arxiv.org/abs/2308.08295) by Tang et al.
>
> [4] [Exploring the limits of domain-adaptive training for detoxifying large-scale language models](https://arxiv.org/abs/2202.04173) by Wang et al.
>
> [5] [TIES-Merging: Resolving interference when merging models](https://openreview.net/forum?id=xtaX3WyCj1) by Yadav et al.

---

> ### Author Response · Authors · 2024-06-07
> **A Gentle Reminder to vKrR**
>
> **Dear Reviewer vKrR,**
>
> As of now, we have not received any responses from the reviewer. If the reviewer has any questions or comments about the rebuttal or the results of the additional experiments, we would be happy to answer them. We apologize for the last-minute reminder during your busy time.

---

### Author Response · Authors · 2024-06-05
**A Gentle Reminder**

**To: Area Chairs and all reviewers**

As of now, we have not received any responses from the reviewers. If any reviewers have any questions or comments about the rebuttal or the results of the additional experiments, we would be happy to answer them. We apologize for the last-minute reminder during your busy time.

---

### Author Response · Authors · 2024-06-07
**Response to Meta Reviewer**

Dear Meta Reviewer,

We have addressed all the reviewers' questions and comments and believe this has resolved all their concerns.
If our paper is accepted, we will include all our responses in the camera-ready version.
The reviewers’ feedback has further enhanced our paper, and we would like to express our gratitude to them once again.

Specifically, we have listed below the answers we provided to the reviewers’ comments and questions.

**Additional Experiment**

We have further conducted the following additional experiments to validate the proposed method's effectiveness and robustness.

- We applied the proposed method to a larger and more recent model, Llama2 - 7B, and observed similar performance as in the experiments with the smaller models (GPT-2, Phi - 1.5). (Please see **[our Rebuttal to Reviewer vKrR](https://openreview.net/forum?id=TBNYjdOazs&noteId=qLvE2RNltY)**)

- We conducted experiments by applying the Ties-Merging [1]  in addition to linear merging which we adopted as a merging method in our approach. The results showed that the proposed method is effective even with the Ties-Merging. (Please refer to **[our Rebuttal to Reviewer vKrR](https://openreview.net/forum?id=TBNYjdOazs&noteId=qLvE2RNltY)**)

- In addition to evaluating model degeneration by perplexity, we further evaluated it on 7 downstream tasks. The results confirmed that models detoxified by the proposed method have lower toxicity while maintaining similar performance in both perplexity and downstream tasks. (Please refer to the table in **[our Rebuttal to Reviewer tRLA](https://openreview.net/forum?id=TBNYjdOazs&noteId=dHnDleniE3)** for the results, and **[our comment to Reviewer tRLA](https://openreview.net/forum?id=TBNYjdOazs&noteId=04JTMxr8VH)** for descriptions of each downstream task)


**Answers to Question**

We have provided the following answers to the questions received from the reviewer.

- Regarding the comparison with existing detoxification methods, we did not compare our proposed method with existing strong self-detoxification approaches [2, 3]. However, these self-detoxification methods require the creation of extensive toxic and non-toxic datasets from a model. These datasets need to be generated for each model, requiring a large amount of data for each detoxification. On the other hand, the task vector negation method adopted in the proposed approach allows for efficient detoxification using only existing toxic datasets. We have confirmed that our proposed method performs better than other detoxification methods that also use only toxic data (i.e., gradient ascent, and the existing task vector negation [4]). (Please refer to Table 1 in our paper)

**Revisions for Writing**

Based on the advice of many reviewers, if our paper is accepted, we plan to make the following changes and additions in the camera-ready version

1. Correct typos and grammatical errors.
2. Clearly describe the purpose and intent of each experiment.
3. Provide more detailed information on the evaluation methods and settings for each experiment (Please refer to **[our Rebuttal to Reviewer tRLA](https://openreview.net/forum?id=TBNYjdOazs&noteId=Tak39hYM8I)**).
4. Include a precise definition of toxicity and examples for each toxicity category (Please also see **[our comment to Reviewer vKrR](https://openreview.net/forum?id=TBNYjdOazs&noteId=X3MKeMdDna)**).

[1] [TIES-Merging: Resolving interference when merging models](https://openreview.net/forum?id=xtaX3WyCj1) by Yadav et al.

[2] [Exploring the limits of domain-adaptive training for detoxifying large-scale language models](https://proceedings.neurips.cc/paper_files/paper/2022/file/e8c20cafe841cba3e31a17488dc9c3f1-Paper-Conference.pdf) by Wang et al.

[3] [CMD: a framework for Context-aware Model self-Detoxification](http://arxiv.org/abs/2308.08295) by Tang et al.

[4] [Editing Models with Task Arithmetic](http://arxiv.org/abs/2212.04089) by Ilharco et al.

---

> ### Comment · Area_Chair_C6kh · 2024-06-07
> **Reminder**
>
> Dear all,
> The discussion period will close soon. Please respond to the authors' rebuttal and let us know if you have any further concerns.

---

### Decision · Program_Chairs · 2024-07-10

**Decision:**

Accept

**Comment:**

The paper presents an interesting approach to detoxification. Most reviewers find this paper well-written, the proposed method is well-motivated, and it provides a great analysis of the proposed approach.

The authors responded to all the concerns and provided additional experiment results to address the comments. However, it is not encouraged to use the discussion mechanism to circumvent the word limit of rebuttal. Also, it's not recommended to add a significant amount of new experiments in the rebuttal as this will drastically change the paper and the paper should go to another round of review.